# Article
# Thematic Presentations in Indonesian Qur'anic Commentaries

Jauhar Azizy [1,*] , Mohammad Anwar Syarifuddin [1,*] and Hani Hilyati Ubaidah [2,*]

1   Faculty of Ushuluddin, Department of Ilmu Al-Qur'an and Tafsir, UIN Syarif Hidayatullah Jakarta, Banten 15412, Indonesia
2   Faculty of Tarbiya and Teachers Training, Department of Islamic Education, UIN Syarif Hidayatullah Jakarta, Banten 15412, Indonesia
*   Correspondence: jauhar.azizy@uinjkt.ac.id (J.A.); anwar.syarifuddin@uinjkt.ac.id (M.A.S.); hani.hilyati@uinjkt.ac.id (H.H.U.)

**Abstract:** The study of thematic interpretation (*tafsīr mawḍū'ī*) in Indonesia focuses primarily on the products of interpretations written in the 2000s, with little attention paid to the origins of thematic interpretations in Indonesia. This article will look at different ways of presenting thematic interpretations in the Indonesian commentary literature prior to the 2000s. This article will also investigate whether the development of the form of interpretation in the Middle East, particularly Egypt, has had any impact on the form of Indonesian thematic interpretation. The methodology used in this study is a literature review based on thematic interpretations (mawḍū'ī) of several Egyptian commentators, including Amīn al-Khūlī (d. 1966), Mahmūd Shaltut (d. 1963), Bint Shāti' (d. 1998), 'Abd al-Hayy al-Farmāwī (d. 2017), Hassan Hanafī (d. 2021), and Mustafā Muslim (d. 2021). The authors also use content analysis to examine some of the Indonesian commentary literature. The conclusion of this article demonstrates that thematic interpretation discourse in Egypt had a significant influence on the development of thematic interpretation in Indonesia, particularly interpretation literature published in the 1990s. This influence can be seen in the presence of a glossary and an index of discussion topics, complete with Qur'anic verses and arranged alphabetically or chronologically. This is in keeping with the spirit of Amīn al-Khūlī (d. 1966), who emphasized the importance of thematic discussions in determining the Qur'anic viewpoint on specific issues.

**Keywords:** methods of interpretation; thematic interpretation; thematic indexes

## 1. Introduction

The methods of Qur'anic interpretation had dealt with humanity's complex problems, which brought forth various methods of interpretation. Muslim scholars have used a variety of methods to interpret the Qur'an. They believe that the Qur'an serves as guidance (*hudan*), and thus it is always relevant. It has been stated that the Qur'an is always in accordance with the circumstances (*ṣālih li kulli zamān wa makān*). 'Abd al-Hayy al-Farmāwī (d. 2017) explored thematic methods of Qur'anic interpretation in his work *Al-Bidayah fī al-Tafsir al-Mawḍū'ī: Dirāsah Manhajiyyah Mawḍū'iyyah* (al-Farmāwī 1977). In addition to the global (*ijmalī*), comprehensive (*tahlīlī*), and comparative (*muqāran*) methods, the thematic method (*Mawḍū'ī*) is defined as a method of interpreting Qur'anic verses according to related themes or subject matters, which connects the purpose of the verse and its thorough comprehension, as well as compiling verses of the Qur'an which have related themes or common direction to gain a general conclusion (al-Farmāwī 1977, pp. 51–52).

If the other methods of interpretation are applied to the chronological interpretation of Qur'anic verses from the beginning to the end of the scripture, the thematic method of interpretation collects Qur'anic verses on the basis of their similarly related themes, in order to discuss their relationship to the others on the basis of the synoptic term (Wielandt 2004). Thematic interpretation is used to (1) interpret a Qur'anic surah by dividing its content into shared specific themes/concerns, and (2) interpret Qur'anic verses on certain predetermined themes and complete discussions by referring the verses to the predetermined themes.

## 2. Result

Thematic interpretation, as emphasized by al-Farmāwī, had been widely used in the Middle East for about a decade prior to the publication of his book. Amīn al-Khūlī (1895–1966) pioneered thematic interpretation in his "study on the surrounding of the Qur'an" (*Dirāsah mā haula al-Qur'ān*) by emphasizing the extrinsic elements of the Qur'an alongside his "study of the Qur'an" (*Dirāsah mā fī al-Qur'ān*) by emphasizing the extrinsic elements of the Qur'an alongside his "study of the Qur'an" (al-Khūlī 1961, pp. 229–39). In fact, he had not organized his studies in a systematic manner, but he emphasized the importance of thematic discussion in order to understand Qur'anic perspectives on specific cases.

Mahmūd Shaltūt (1893–1963) expanded on al-Khūlī's idea in 1960 with his *Tafsīr al-Qurān al-Karim: Al-'Ajza' al-'Ashrah al-Ūlā* ("The Interpretation of the Qur'an: The First Ten Divisions") (Shaltūt 1960). Shaltūt was a professor at the Faculty of Theology, al-Azhar University. By focusing forms of interpretation on key ideas, he provided a middle ground between the chronological and thematic approaches (Jansen 1974, p. 14 in Wielandt 2004, p. 62). Shaltūt, like al-Khūlī, had not comprehensively formulated steps of thematic interpretations. In fact, al-Khūlī's idea was also proposed by his wife, 'Aisha 'Abd al-Rahmān, best known by her pen name Bint al-Shāti' (1913–1998). She wrote *Al-Tafsīr al-Bayānī li al-Qur'ān*, which translates as "the stylistic interpretation of the Qur'an" (Shāti' 1962). She compiled Qur'anic verses based on specific keywords. Bint al-Shāti' (d. 1998) clearly applied al-Khūlī's theory in practice to certain surahs of the Qur'an. He used surahs or parts of surahs from the Qur'an as well as the occasion of revelation (*asbāb al-nuzul*) in her analyses (Wielandt 2004, p. 75). However, such a collection of verses could not be considered a thematic study at this time.

Around a decade later, Ahmad Sayyid al-Kūmī (b. 1912) attempted to introduce thematic interpretation as one of the subjects taught at al-Azhar University. Al-Farmāwī (d. 2017), one of al-Kūmī's disciples, developed the methodological approaches to thematic interpretation in his book *al-Bidāyah fī al-Tafsīr al- mawḍū'ī: Dirasāh Manhajiyyah mawḍū'īyyah*, "The Beginning of Thematic Interpretation: A Methodologically Thematic Study" (al-Farmāwī 1977). Al-Farmāwī's methodological formula was then completed by his colleague 'Abd al-Sattar Fath Allāh in his book *Al-Madkhal ilā al-Tafsīr al-Mawḍū'ī*, "The Introduction to Thematic Interpretation" (Fath Allāh 1991). Fath Allāh emphasizes that the exegete's predetermined themes must have been indicated within the redaction of the Qur'anic text.

Mustafā Muslim introduced a form of inter-surah or inter-verses discussion on thematic method of interpretation in his *Mabāhith fī al-Tafsīr al-Mawḍū'ī*, "Studies on Thematic Interpretation" (Muslim 1989). At the same time, Hassan Hanafī (d. 2021) emphasizes the importance of dialog between the exegete (who scrutinizes problems he encounters), the text, and the context of the Qur'anic verses in his book *al-Dīn wa al-Thawrah*, "Religion and Revolution" (Hanafī 1989). An interpretation, according to him, is a manifested form of the exegete's social position within his social structure. As a result, an interpretation must be applied in practice rather than theoretically. Salāh 'Abd al-Fattāh al-Khālidī (b. 1947) continued a similar idea in his book *Al-Tafsīr al-Mawḍū'ī baina al-Naẓariyyah wa al-Tatbīq*, "Thematic Interpretation in between Theory and Practice" (Khālidī 1997).

The trend of thematic interpretation did not emerge solely in Egypt. It is spreading throughout the Islamic world. Muhammad Quraish Shihab was a pivotal figure in the introduction of al-Farmāwī's thematic method of interpretation in South-East Asia prior to the millennium's turn. Shihab has always referred to al-Farmāwī in his thematic discussions. Prior to the publication of *Membumikan al-Qur'an*, "Grounding the Qur'an," in 1992 (Shihab 1992) and *Wawasan al-Qur'an: Tafsir Maudhu'i atas Pelbagai Persoalan umat*, "The Qur'an's Insights: Maudhu'i's of Various People's Problems," in 1996 (Shihab 1996), there had been several studies on the Qur'an based on thematic discussion, such as Bahroem Rangkuti (Rangkuti 1960); M. Said (Said 1960); Mustafa Baisa (Baisa 1960); Hasri (Hasri 1969); Bey Arifin (Arifin 1972); and Harifudin Cawidu (Cawidu 1989). According to these studies, thematic interpretation discourse was quite popular in Indonesia at the time. Since the 1960s, there have been several widespread patterns of thematic interpretation, namely (1) interpreting one or a collection

of surahs to be assigned themes and (2) determining predestined themes and then collecting the related Qur'anic verses dealing with those themes (Federspiel 1996a; Gusmian 2003; Baidan 2003).

In Indonesia, there is still a scarcity of research on early forms of thematic interpretation. This essay investigates discourses on several models of thematic interpretation developed in the Middle East, particularly Egypt, to the emergence of such an awareness applied in forms of ways of presenting thematic discussion in tafsir literature. Some examples include presenting table of contents and thematic indexes of Qur'anic verses within chronological commentaries of the Qur'an. Such methods have been used in several new tafsir publications, as well as previous tafsir works to be reprinted in new editions, along with the revised spelling of Bahasa Indonesia. The appearance of such new tafsir editions leads to the conclusion that the introduction of thematic interpretation in the Middle East has significantly influenced the growing interest among Indonesian exegetes to make some types of adaptation at the earliest level by presenting thematic indexes to their renewed tafsirs.

Various changes are being made in the process of republishing those commentaries. Among the considerations is that their books should meet the new expectations of their readers in order for them to easily access the contents of the commentaries. By including some forms of thematic indexes in the new publication of certain brief tafsirs such as *al-Furqān* (by Hassan 2010) and *al-Bayan* (by Ash-Shiddieqy 2012), readers of those tafsirs can still have faster access to the contents of the tafsir. The readers are not required to read the entire text of the books, which have a minimum of a thousand pages. With the introduction of the thematic method of interpretation to the Qur'an, accessing tafsir literature without the table of contents or thematic indexes was deemed difficult for their readers in quickly finding any information they required. As a result, if the authors had not changed the way they presented their books, works of chronological tafsir might have been discarded because people considered those tafsirs to be out of date. Thus, even in its early stages, the introduction of thematic interpretation has undoubtedly aided the rise of a new pattern of writing Qur'anic commentaries.

There are some forms of thematic presentations that can be accommodated within the attempts to republish some tafsir books. The addition of titles or themes that were the subject matters for the series of interpreted verses was an early effort that was noticed. Readers can simply read the essence of the collective verses via a detailed sub-title indicating the verses' generic content. Since the early arrangements of *Al-Qur'an dan Terjemahnya* by the Team of the Ministry of Religious Affairs, such an effort has been made (1965). Because of this, in fact, it is a book of translation of the Qur'an; however, it will be difficult to classify this book as *tafsīr* because the same team has begun to write a more comprehensive project of *al-Qur'an dan Tafsirnya*. As a result, the following discussion does not include this translation book.

However, such a pioneering effort was imitated in a far more innovative form by exegetes in subsequent periods. The introduction of thematic interpretation in Egypt had a significant impact on the rise of a somewhat similar concern within the development of Indonesian tafsir literature, namely the addition of tables of contents, thematic indexes, glossaries, and references to Qur'anic verses. Furthermore, they provided page numbers where the verses or explanations were found in their books. The following tafsirs have tables of contents and thematic indexes:

1. *Tafsir Qur'an Karim* by Mahmud Yunus, first published in the 1950s and reprinted in 1973 with the advanced spelling of Indonesian.
2. Bachtiar Surin's Terjemah dan Tafsir al-Qur'an 30 Juz Huruf Arab dan Latin, published in 1976.
3. *Tafsir Rahmat* by Oemar Bakry, who first published his works in 1981.
4. *Tafsir al-Furqan* by Ahmad Hassan, which was first published in 1956 and was reprinted in 2005, with a new edition published in 2010.

5.  *Tafsir al-Bayan* by Hasbi ash-Siddieqy, first published in 1966 (Ash-Shiddieqy 1966), republished in 2002, and reprinted in 2012.

## 3. Discussion

### 3.1. Tafsir Qur'an Karim by Mahmud Yunus (1899–1982)

Mahmud Yunus spearheaded the first Qur'anic translation into Indonesian. His translation of the Qur'an sparked the creation of similar works, including some extensive commentaries in the 1960s. His book's thematic index of Qur'anic scientific verses had become the standard reference for other tafsirs. Mahmud Yunus was born on 10 February 1899, in Batusangkar, West Sumatra. His father, Yunus bin Incek, was a village imam. His mother was a granddaughter of Sheikh Muhammad Ali, known as Tuanku Kolok, and his maternal uncle, H. Ibrahim Dt. Sinaro Sati, was a wealthy trader who played an important role in supporting Yunus' higher education in Egypt (Rina 2011, pp. 170–71). Yunus grew up in an Islamic education system known as surau. First, he learned Qur'an recitation from Muhammad Tahir bin Muhammad Ali, also known as Engku Gadang. He attended the Elementary School until grade three, when he transferred to the Madras School in Tanjung Pauh, which was run by HM Thaib Umar. In addition to being a student, he taught his junior fellow there for about 8 years. He was finally appointed as the school's headmaster in 1917.

His introduction to the reform movement of Muhammad 'Abduh (d. 1905) and Rashīd Ridā (d. 1935) came via the *al-Manar* magazine, which fueled his desire to continue his studies in Egypt. Following the failure of his first attempt in 1920, he received a visa from the British Colonial Government in 1923, allowing him to travel from Penang to Saudi Arabia and then to Egypt. He was admitted to al-Azhar University in Cairo in 1924, where he studied Islamic Jurisprudence, Tafsīr, and the Hanafi School of Islamic Law. In 1925, he received his diploma from al-Azhar. He then continued his education at Madrasah Dār al-'Ulūm 'Ulyā, where he earned a diploma in Islamic education in 1930. He went back to Batusangkar, his hometown. His appointment as the first rector of ADIA Djakarta in 1957, then as the dean of the faculty of Islamic education at IAIN Syarif Hidayatullah Jakarta, was among his major achievements in his academic career in 1960.

*Tafsir Qur'an Karim* was first published in 1950. *Tafsir Qur'an Karim* is the culmination of Yunus' lifetime efforts in translating and interpreting the Qur'an from 1922 to 1973, as well as the growing development of the Indonesian language. Mahmud Yunus' commentary is divided into two parts. On the right side, he wrote the verses of the Qur'an in Arabic script, and on the left, he wrote the translation. In some cases, he included annotated commentary in the form of additional explanations for verses that needed more information. This commentary was placed at the bottom of the page, similar to footnotes. The translation does not take up more than a half-page. In the case of a lengthy commentary, he would prefer to continue his explanation on the following pages in a chronological order. In short, Yunus presented the Mushaf in chronological order, verse by verse and surah by surah. Yusuf's commentary is brief in nature, or we can say that Yunus used the global (*ijmalī*) method of interpretation. These brief commentaries, on the other hand, include narrations of the times of revelation (*asbāb al-nuzūl*). Mahmud Yunus's pattern of interpretation combined traditional (*bi al-ma'thūr*) and rational (*bi al-ra'y*) interpretations

Concerning the sources of his interpretation, Yunus did not provide a list of tafsīr he had referred to for his interpretation, but he agreed with fellow Muslim exegetes such as Ṭabarī, Ibn Kathīr, and al-Qāsimī that the core hermeneutical procedures of Qur'anic interpretation are primarily based on *Qur'ān bi al-Qur'ān* explication. In this context, the interpretation of a Qur'anic verse should essentially refer to any explanation mentioned in other parts of the Qur'an. If such an inter-Qur'an explanation cannot be found, one must refer to sound prophetic hadiths. If there are no traditional explanations originating from the Prophet, one can take sequential accounts of the Prophet's companions (*ṣaḥāba*), as well as their followers (*tābi'ūn*). Yunus, on the other hand, limited himself to taking companions' accounts only on occasions of revelation, not their personal views on tafsīr. According to

him, followers' accounts can be used as a source of tafsir as long as their statements are in accordance with Muslim convention *(ijmā')*. He added several other principles to the above-mentioned bases of hermeneutical procedures: (1) an interpretation based on general comprehension of the Arabic lexicon, (2) an interpretation based on personal examination *(ijtihād)* by jurists, and (3) a rational interpretation by the Mu'tazilites (Yunus 1983, p. vii).

Yunus has not given any titles to his translations or commentaries on specific verses. He only provided a marker, such as "description of the verse . . . page . . . " Yunus, on the other hand, provided a table of contents that listed the names of the surahs as well as the contents of the given commentary. There is no mention of verse numbers, as there are only the titles or conclusions of the given interpretation, as well as the page numbers. Each title generally represents an interpretation of a single verse. If there are many verses to be interpreted on a single page, the page will be referred to with many headings based on the contents of the interpreted verses. As a result, there are several types of tables of contents, which discontinue the page number in favor of separated numbers in roman numerals at the top center.

In the 1973 reprint edition, Mahmud Yunus appears to add some new things to the *Tafsir Qur'an Karim* in terms of thematic presentation, concerning not only the contents of his comprised commentary but also what he refers to as the "Conclusion of the Qur'an". The page numbers are given in roman numerals at the bottom center. As a result, the page number differs from the table of contents, which was previously mentioned. This section serves as an additional thematic index, presenting major themes of the Qur'an as well as references to related verses, complete with page numbers in the book. When dealing with the theme "Faith in God", for example, Mahmud Yunus provided comprehension of the theme through references to verse numbers, surah numbers, and the page number of the text in his book.

The titles in the *Tafsir Qur'an Karim's* thematic index include not only related themes on Islamic teachings, such as theological and legal aspects of Islam, but also related themes on economy, the relationship of the Qur'an with sciences, history, and social problems. Even though the trend of thematic commentary was not thriving when the book was reprinted in 1973, it appears that Yunus was heavily influenced by the ideas of Egyptian thinkers emphasizing the need for thematic commentary expressed by Amīn al-Khūlī (d. 1966) and Bint al-Shāti' (d. 1998) by the end of the 1960s. It appears that Yunus's status as an al-Azhar alumnus provided him with strong access to the introduction of thematic commentary within the development of Qur'anic studies. The thematic index presented by Mahmud Yunus is not a new model of thematic commentary, but it can be viewed as an embryo highlighting the ongoing current of thematic interpretation methods for the coming decades.

### 3.2. Terjemah & Tafsir al-Qur'an by Bachtiar Surin

Bachtiar Surin's Qur'anic translation and commentary were published in the mid-1970s, shortly after Mahmud Yunus re-published his exegesis in 1973. In his introduction, he states that the tafsir was completed on 11 September 1976. Nonetheless, the year of publication of this commentary was 1978, as it followed the issuance of a letter of correction *(taṣḥīh)* categorized as "special edition of *muṣhaf* with Latin transliteration" from the Indonesian Ministry of Religious Affairs' Lajnah Pentashhih Mushaf al-Qur'an. There are some prefaces for the publication requested from various personages that would have made this new form of tafsir be published almost two years after its completion.

Everyone seems to agree that this interpretation is an essential work of Qur'anic translation and tafsir. Hamka, the chairman of MUI (the Indonesian Ulama Assembly), emphasized that the book of the Qur'an with Latin transliteration would benefit those who want to understand the content of the Qur'an but find it difficult to read the Qur'an in Arabic. Meanwhile, the concurrent Minister of Religious Affairs, Alamsjah Ratu Perwirane-gara, agreed that efforts to publish translations of the Qur'an as well as interpretations in

both Arabic and Latin scripts had made it easier for those who wanted to understand the Qur'an's language. As a result, he believes that such an attempt deserves to be recognized.

In his preface, Minister of Home Affairs Amir Machmud agreed with the benefits derived from the publication, which was expected to increase worship activities in Indonesia and build national insight. For him, the tafsir writer's efforts are far-reaching endeavors, allowing for an easier and deeper understanding for those who still struggle to read the Qur'an in its original letter and language. He hoped that by making such an effort, more people would be able to find the spirit of Islam in order to anticipate the destructive effects of development through the guide of divine light. Based on the preceding statements, it is reasonable to conclude that the effort of translating the Qur'an, as well as its interpretation, is part of the larger effort to develop spirituality. In his remarks, Bachtiar Surin, the director of Firma Sumatra, insisted that if the teachings of the Qur'an can be carried out purely and earnestly through this translation, the effort can be the seeds of rapid advancement as well as embody spiritual development within the surrounding regions (Surin 1978, pp. vii–xv).

The translation is the result of a collaborative effort led by Bachtiar Surin and two members, M. Said and Zainuddin Sulaiman, on the order of Firma Sumatra (often abbreviated as Fa Sumatra), led by H. Bahar Surin. Fa Sumatra is a publishing company based in Bandung. The dedication of the book for the sake of "Our deceased parent H. Surin, who died in West Sumatra in 1926" would indicate that H. Bahar Surin was a sibling of Bahtiar Surin, according to a note on the book's cover. According to the statement, both H. Bahar Surin and Bachtiar Surin are West Sumatran natives who settled in Bandung and built the publishing industry there.

*Terjemah & Tafsir al-Qur'an 30 Juz Huruf Arab dan Latin* was the first edition of this book. This work is comprised of 1462 pages in one volume, plus 52 pages of introduction. It was published by Fa Sumatra, a Bandung-based publisher. As a writer, only Bachtiar Surin's name appears on both the front and inside covers. According to the book's introduction, he was the chairman and person in charge of a team called Lembaga Penerjemah Kitab Suci al-Qur'an. H. Bahar Surin, the director of the Fa Sumatra, had established the working standards in the introduction. Among them is the requirement to display "clear pictures" in Qur'anic verse translations.

A literal translation is permitted and preferred if it is capable of presenting the clarity of the verse's intent. If there is an irregularity in its literal interpretation, the translation effort is carried out through an explanation by adding several additional words, so that the understanding can be directed toward more clarity, and thus the resultant translation can be more prominent. In other words, in addition to attempting literal translation, this work offers a rather meaningful translation for verses that were too enigmatic to be understood by its literal meaning.

The debate will be loaded, if necessary, in terms of explanatory sentences that form a meaningful translation of the Qur'anic verses among various types of controversies. However, the book will only display the strongest opinion with a convincing proposition based on the most powerful meaning. Furthermore, if there are differences in rules in both Arabic and Indonesian, the team was asked to avoid linguistic abnormalities, as reader boredom would be contrary to the Prophet's advice (Surin 1978, pp. xvii–xviii). Concerning the reasons for this new effort to provide a method of reciting the Qur'an in both Arabic and Latin scripts, it was clearly stated that the included Latin script is not a replacement for the original Arabic letters. It is simply part of the effort to assist those who do not know Arabic letters in continuing to read the Qur'an.

One of the principles held by this institution is that it is preferable to read the Qur'an by adjusting one's possessed ability, even by reading its Latin script, rather than not reading the Qur'an at all (Surin 1978, p. xviii). Meanwhile, the standard transcription used in this book appears to follow Ministry of Religious Affairs guidelines that apply the proximate sound of the Arabic origin to its Latin equivalent. For letters that cannot be matched to their respective counterparts, a double consonant transliteration pattern with an underscore will be used, such as dh, th, zh, gh, sy, sh, ts, kh, and dz. Because there is no other option, the

letters hā' and hā 'are all written without divergent diacritical marks in this transliteration system, highlighting the underlying weakness of this early transliteration form. Long vowel letters, on the other hand, are only written with the same vowel letter doubled to indicate the reading of two harakat. Longer tones are denoted by a diacritical mark in the double vowel, as in *waladh dhāal-liin*.

This book's sources of interpretation range from classical to medieval to modern commentaries. Only the Tafsir at-abar is mentioned among the classical tafsirs, while medieval commentaries include interpretations from Ibn Kathīr, al-Baiḍāwī, Zamakhshari, Nasafi, Jalal al-Din al-Suyutī, and Abu Sa'ūd. Finally, among the modern tafsirs used in this book are the the Tafsirs *al-Manar, Jawāhir*, *al-Maraghī, fī Ẓilāl al-Qur'an*, and the English translation *The Holy Qur'an* by Abdullah Yusuf Ali. Meanwhile, almost all Nusantara tafsirs published before the last quarter of the twentieth century are used as references and interpretation sources for this book. In addition to exegesis books, this work also refers to books in the fields of Islamic jurisprudence, Arabic language, and other general literature (Surin 1978, pp. xxiii–xxiv). Based on the sources used, it is possible to conclude that the authors' approach brings a mixed style of interpretation between the traditional approach and the rational approach of interpretations.

In the back of the book, there are several tables of contents. The first is a list of the parts (juz'), which describes which surah and verse numbers the juz' begins with and which surah and verse numbers it ends with, as well as the subsequent page numbers in the book. The list is organized chronologically, beginning at the top and ending at the bottom. In addition, this book includes a list of the *surahs*. The names of the surah are sorted according to their order in the Qur'an and are accompanied by the translations/meanings of the surah, the total number of its verses, and the page on which it is located in the book. The list concludes with a prayer recited after finishing the Qur'an (Surin 1978, pp. xxvii–xxix).

The translation of surah names is completed both literally and metaphorically. According to the standard principle of translation established by the publisher, the literal translation of surah names is seen as the main reference, whereas meaningful translation is seen in several cases, such as when translating the meaning of the surah *al-Taubah*, which was interpreted as "termination of relations" (*pemutusan hubungan*), *al-Muzammil* was translated as "orang berkelumun" (being covered), and *al-Muddaththir* was translated as "orang yang berselimut" (a person covered with a blanket). Differences in the last two-surah meanings may be necessary to avoid confusion, given that the names of the two surahs are similar in meaning and are located sequentially. Meanwhile, events that mark the events of the doomsday are referred to by various terms.

The meaning of "Judgment Day" is given only to the surahs of *al-Hāqqah* and *al-Qiyāmah*, whereas other similar terms are defined according to their literal comprehension, so that the name *al-Wāqi'ah* is interpreted as "a terrible event" (Kejadian yang dahsyat) and *al-Qāri'ah* is interpreted as "a thrilling catastrophe" (malapetaka yang mendebarkan hati). The last two actions appear to be taken to avoid confusion. While the preference for literal translation also appears in interpreting some surah names that mark the phenomenon of destruction during the course of the apocalypse, such as *al-Takwīr* was interpreted "to roll", *al-Infitār* means "falling apart" (gugur berantakan), *al-Inshiqāq* means "ruined" (porak poranda), *al-Gāshiyah* as "the day of the catastrophe" (hari selubung malapetaka), al-*Zalzalah* (goncangan yang dahsyat).

The second list is of titles that are made up of a verse or a collection of verses. This list denotes thematic titles of the translated content of a verse or several groups of verses, and it was also made alphabetically by the first letter of its sentence or phrase, rather than by the theme. In contrast to the titles listed in *Al-Qur'an dan Terjemahnya* by the Ministry of Religious Affairs team, which were arranged according to the titles or themes along the chronology of the Mushaf, the motivation for listing the thematic indexes in alphabetical order may be similar to avoid confusion. This table of contents is less useful at first glance because not all title-marker-initials are keywords in the presented theme, so the title *Bila datangnya hari kiamat* (lit. when does the doomsday happen) is classified into the letter *B* for

"bila" rather than the letter *K* for "kiamat". Such would have been pointless because *bila* is not a unique keyword.

Thematic presentations of the table of contents, as well as thematic titles arranged by Bachtiar Surin in his commentary, demonstrate the influence of the development of thematic method of interpretation of the Qur'an. It is similar to the indexing found in previous tafsir and translation of the Qur'an publications, such as *al-Qur'an dan Terjemahnya* by the Ministry of Religious Affairs team (1965) and *Tafsir Qur'an Karim* by Mahmud Yunus (1983). The table of contents and thematic indexes were designed to make it easy for readers to explore the meaning of the Qur'an within the book. Indeed, the table of contents and thematic indexes that reflect the contents of the Qur'an verses are not similar to those of Mahmud Yunus, because the list and thematic indexes compiled by Bachtiar Surin are more similar to the composition of the list of titles of the Ministry of Religious Affairs translation (1965).

Bachtiar Surin, on the other hand, went a step further by not presenting the index chronologically, as the Kemenag Team had, but by sorting the titles alphabetically. Nonetheless, it is less meaningful because the index was still based on the first letter of the sentences or phrases, rather than the first letters of the formulated keywords. After all, such an effort is still a form of innovation that should be lauded. Such flaws may have prompted corrections and reprinting for the book's next publication. Similarly, it may spark new approaches to publishing similar tafsir books in the near future.

*3.3. Tafsir Rahmat by Oemar Bakry*

*Tafsir Rahmat* by Oemar Bakry was written in the early 1980s, when ideas about thematic interpretation were becoming quite common, particularly in Egypt and the Middle East in general. This can be considered the formative period of thematic interpretation. Through communication networks intertwined among the alumni of Middle Eastern universities, particularly the Azhar, the new theory began to spread more evenly throughout the Islamic lands. The dissemination of ideas on thematic interpretation would have been more vibrant in the early 1980s compared to a decade earlier when Mahmud Yunus republished *Tafsir Qur'an Karim* in 1973. As a result of this, more development and systematic indexing have begun to appear in tafsir literature, such as Oemar Bakry's *Tafsir Rahmat*.

Oemar Bakry was born on 26 June 1916 in a small village called Kacang on the outskirts of Lake Singkarak, Solok, West Sumatra (Bakry 1983). Bakry attended Sekolah Sambungan in Singkarak after finishing elementary school in his village. Following that, he enrolled at the Diniyah Putra in Padang Panjang and graduated in 1931. The following year, in 1932, he graduated from Thawalib School. Bakry continued his scientific journey to Padang from Padang Panjang. He enrolled in the Kuliyatul Mu'alimin al-Islamiyah and graduated in 1936. In 1954, eighteen years after graduating from Kuliyatul Mu'alimin in Padang, he tried his luck in Jakarta by enrolling at the Faculty of Letters University of Indonesia. His higher education at the University of Indonesia, on the other hand, was never completed. There was no explanation for the causes and business that prevented him from finishing his college studies at University of Indonesia. Because of his publishing business, which took up most of his time, he was unable to complete his studies at the Faculty of Letters, University of Indonesia.

His educational career began when he began working as an active teacher in various schools. From 1933 to 1936, he taught at Thawalib School in Padang Panjang. He was appointed director of the Muhammadiyah School of Padang Sidempuan, North Sumatra, a year later, in 1937. After only a year in Padang Sidempuan, he returned to Padang Panjang and taught at Thawalib School until the Japanese occupation forces arrived in 1938. Simultaneously, he was appointed director of The Public Typewriting School, which was established on 21 January 1938 in Padang Panjang. This latter school was renamed The Garden of Progress (*Taman Kemajuan*) and is still in operation today.

A career in typing school, presumably, is what propelled him into the publishing world, as he travels to Java and manages several publishing houses in Jakarta and Bandung.

In addition to teaching during his stay in Padang Panjang, he is also politically active. He became a member of several social and political organizations, including Permi in the 1930s and the Masyumi Party of Central Sumatra, which encompassed West Sumatra, Riau, and Jambi. After relocating to Jakarta, Bakry served as the chairman of the Indonesian Publishers Association (IKAPI) for the Greater Jakarta area for several years. In Jakarta, he is also the chairman of the Al-Falah Foundation, the Nobel Qur'an Preservation Foundation (Yayasan Pemelihara al-Qur'anul Karim), and the Thawalib Foundation

Bakry was a successful entrepreneur in the printing and publishing industry as well as a former teacher who was active in various social organizations. He is the founder and President Director of "Mutiara" Offset Publishing and Printing in Jakarta, as well as "Angkasa" Publisher in Bandung. On 1 November 1951, he established "Mutiara" in Bukittinggi, and in 1972, he established the same offset printing in Jakarta. Meanwhile, the Angkasa publishing house was established in Bandung on 13 January 1966. Because of his perseverance in the printing and publishing business, he was able to collaborate with overseas publishers, and he was frequently able to attend meetings at both the national and international levels.

He attended the International Publisher Association (IPA) congress in Kyoto, Japan, in 1976, and the same event in Copenhagen, Denmark, in 1980. In addition, he was active in the Islamic propagation movement (*da'wah*), which took him to many parts of the archipelago, including Egypt, where he was invited to give a lecture at al-Azhar University in Cairo in 1983. He also lectured and lectured at various religious universities in his homeland, including IAIN Sunan Ampel Surabaya (11 February 1984), University of Bung Hatta Padang (28 March 1984), and IAIN Imam Bonjol Padang (26 March 1986).

*Tafsir Rahmat* was first published in 1981 in Jakarta. The date was derived from the preface to H. Oemar Bakry's first printed publication (Bakry 1983, p. xvii). This commentary is published in a single volume. Because the compilation method employs the global method of interpretation, it is both concise and dense. This book was printed in a single volume for a variety of reasons. It simply mimics the publication of Tafsir *al-Muṣḥaf al-Mufassar* by Egyptian exegete Muhammad Farīd Wajdī (d. 1954). One of the reasons is to make it easier for readers who do not have a lot of free time to read the Qur'an to follow its guidance without having to open lengthy interpretations. Furthermore, the description is "solid" and "proper," as it is free of controversial issues or israiliyyat stories. It is expected that readers will be able to decipher the meanings of the Qur'anic verses as life guidelines (Bakry 1983, p. xvi).

His interpretation is based on reference books of reputed exegesis such as *Tafsīr al-Manār*, *Tafsīr al-Marāghī*, *al-Tafsīr al-Farīd fī al-Qur'ān al-Majīd* by Muhammad 'Abd al-Mu'īn al-Jamāl, *Tafsīr Ibn Kathīr*, *Tafsīr Fī Ẓilāl al-Qur'ān*, and some previous works by Indonesian exegetes like the translations of the Indonesian Ministry of Religious Affairs, Mahmud Yunus, Zainuddin Hamidi, and Hasbi Ash-Shiddieqy (Bakry 1983, pp. xiv–xv). Based on the book's references, we can conclude that this Qur'an commentary is a hybrid of traditional (*naqli*) and rational (*ra'y*) interpretations.

"The Source of Da'wah" is the title of a thematic index that serves as an additional supplement to *Tafsir Rahmat's* book. Bakry compiled approximately 145 themes that he referred to as "Islamic propagation mottos," which he believes preachers can use in carrying out their duties. He classified 145 mottos into 10 major categories: (1) al-Qur'an, (2) Faith, (3) Worship, (4) Marriage, (5) Science and Technology, (6) Health, (7) Economics, (8) Society and State, (9) Noble Characters, and (10) History. The author creates smaller sub-themes from each of these major themes, and each sub-theme is then provided with the related verses of the Qur'an by mentioning surah names, verse numbers, surah numbers, and the numbers of referential pages within the book (Bakry 1983, pp. 1273–311).

Such a claim reminds us of the same index developed around a decade ago by Mahmud Yunus. The index contains nearly identical composite themes, with a few tweaks for the latter. This brings to mind Bakry's use of the *Tafsir Qur'an Karim* in the composition of his tafsir. Unfortunately, because this index is not alphabetical, the classification is based

solely on the ten major themes. Some examples of the contents of this thematic index are: *Firstly* (1) the Qur'an becomes a grace and guidance for men in QS. 17: 82; 17: 9; 27: 1–2; 17: 89; and 20: 2. (2) In Arabic, the Qur'an is descended in QS. 20: 113; 12: 2; 39: 28; 41: 3; 42: 7; 43: 3; 26: 195. (3) The contents of the Qur'an, which is derived in Arabic, should be understood in QS. 47: 24; 12: 2; 4: 82; 39: 27; 43: 3. (4) The Qur'an provides guidance and mercy to believers in QS. 52, 7: 203, 10: 57, 12: 111, 16: 64, 16: 89, 17: 82, and 27: 77 (Bakry 1983, p. 1275). According to the examples above, the index is still arranged randomly, as is the reference to Qur'anic verses. The reference of Qur'anic verses from each sub-theme is largely not compiled entirely in chronological order based on the order of the number of surahs in the mushaf. In fact, the titles of the sub-themes appear to be repetitive, despite references to slightly different Qur'anic verses.

Bakry compiled an index of prayers, orders, and prohibitions in addition to the thematic index. The titles of the themes of the prayers/orders/prohibitions are then given their reference of the names and numbers of both the surahs and verses, as well as the page location in the book. The classification order is sorted according to the mushaf's chronology, beginning with the prayer in Surah al-Fātihah, then the orders, prohibitions, and prayers in Surah al-Baqarah, and so on until the end of the Qur'an. Oemar Bakry lists two types of tables of contents at the end of his book, one based on the chronological order of the mushaf and the other on an alphabetic index of surah names. The thematic index and tables of contents of the surah, including the list of prohibitions and orders contained in the Qur'an that are listed on the back of the book, can be regarded as an attempt to adopt the development of thematic method of interpretation, despite being compiled in a very simple form of presentation.

### 3.4. Tafsir Al-Bayan by Hasbi Ash Shiddieqy (1904–1975)

Muhammad Hasbi Ash-Shiddieqy was born on 10 March 1904 in Lhokseumawe, North Aceh. He was sent to Tengku Chik in Piyeung to study Arabic for a year when he was 12 years old. He then relocated to Blukbayu to Tengku Chik's dayah. After a year, he relocated to Tengku Chik in Blangkabu Geudong. After only a year, he moved to the dayah of Tengku Chik in Blang Manyak Samakurok. In 1916, he went on to Tengku Chik in Tanjung Barat, i.e., Idris Samalanga, the owner of the greatest dayah in North Aceh for the study of Islamic jurisprudence. Hasbi studied here for two years before transferring to Tengku Chik Hasan in Kruengkale. Tengku Hasbi then obtained a diploma (*shahadah*) as proof of his knowledge, which had reached sufficient levels for him to open his own dayah. He then went back to Lhokseumawe (Nouruzzaman 1977, pp. 13–14). When Hasbi returned to his hometown, he met Sheikh al-Kalali, who advised him to go to Surabaya to study at the al-Irsyad College, which Ahmad Surkati founded in 1926. Hasbi was escorted by Shaykh al-Kalali and was able to enter the specialist level (*takhaṣuṣ*) for a year and a half in order to deepen his Arabic. Hasbi's *takhassus* education at al-Irsyad was his last formal education; since then, he had only read books (Nouruzzaman 1977, pp. 15–16).

After attending the 15th Indonesian Muslim Congress in Yogyakarta in 1949, Minister of Religious Affairs KH Wahid Hasyim offered Hasbi a position as a lecturer at PTAIN, promoting hadith subjects. His expertise in hadith did not dampen his interest in Islamic law from the start, and he was promoted to professor with an inaugural lecture titled "Islamic Sharia Responding to the Challenges of the Age". This inaugural speech commemorated the anniversary of the transition of PTAIN to IAIN on 2 Rabiul Awwal 1381/1961 in Yogyakarta. Hasbi died on 9 December 1975, and his works were read by Malay-speaking Muslims living not only in Indonesia but also in Malaysia and Singapore.

*Tafsir al-Bayan* was first published in 1966, after Hasbi finished a translation of an-Nur in 1961. Hasbi's interpretation of an-Nur is one of the first commentaries he completed since his arrival in Yogyakarta in 1951. *Tafsir al-Bayan* himself completed it in 1966 as a global interpretive work that differs methodologically from the interpretation of *an-Nur*. He finished the eight-volume *Mutiara Hadith* in 1968, and the text of the Collection of the Hadith of the Law in 1971 (11 volumes, but only 6 volumes were published)

([Nouruzzaman 1977](#), pp. 2765–81). This section goes over Hasbi Ash-Shiddieqy's life and his exegetical work *Tafsir al-Bayan*.

*Tafsir al-Bayan* is a commentary that is organized globally. The verses of the Qur'an are written in Arabic in the Madinan muṣḥaf chronological order, with each end of the page ending with verse end. This commentary is organized globally because, in addition to presenting the translated text around its Arabic text, the exegete's additional explanation is brief and very succinct. Hasbi's annotated translation took up only a quarter of a page. The unfilled columns are intentionally left empty on some pages where he did not provide any additional information due to the clarity of the verse. In his commentary, there is no title for any additional explanation. The titles are arranged chronologically, not alphabetically, based on the chronology of the book. By looking at the contents, the reader will be able to identify the contents based on the presented theme. In contrast to Mahmud Yunus' *Tafsir Qur'an Karim*, which places all of the contents and indexes at the back of the book, the tables of contents are placed in the front.

*Tafsir al-Bayan* accommodates the development of thematic method interpretation by including an additional thematic index called "glossary" with a series of pages that merge and intertwine with the tafsir's text. This glossary is based on Arabic terms that do not distinguish the letters *alif* from *'ain*. In the glossary, definitions of terms are provided, as well as references to the number of verses related to similar discussions in the Qur'an. However, it excludes the reference to the related discussion in the book in this book. An example of the displayed contents of the book's thematic index concerning the term *khalq* was interpreted in several ways, among them: (1) forming the form, as stated in QS. 5: 110, (2) predicting, as stated in several verses (QS. 29: 17, (3) creating something and starting its creation (QS. 7: 18; (4) predestination, and (5) religion of Allah, character, and nature (QS. 30: 40) ([Ash-Shiddieqy 2012](#), p. 610). Although some surahs are included in its comprehensive thematic index, some terms are only presented by their meaning, with no references to Qur'anic verses. *Tafsir Bayan* by Hasbi Ash-Shiddieqy was republished in a new form in 2002 and 2012 as a result of Hasbi's son, the H.Z. Fuad Hasbi Ash-Shiddieqy edition. Because thematic presentation of the book was so simple, it developed only slightly.

### 3.5. Al-Furqan Tafsir Qur'an by Ahmad Hassan (1887–1958)

Ahmad Hassan was born in 1887 in Singapore. At the age of seven, he began studying the Qur'an and then enrolled in a Malay school. His father was eager for Hassan to learn a variety of languages, including Arabic, English, Malay, and Tamil ([Hassan 2001](#), p. ii). He spent about three years learning from Sa'īd 'Abdullah al-Musāwī, then Sheikh Hassan al-Malābarī and Sheikh Ibrāhīm al-Hindī. It was finished around 1910. Hassan studied Arabic until he was 23 years old because it helped him understand other sciences such as tafsir, fiqh, farā'id, and mantiq ([Mughni 1984](#), p. 13). In 1911, Hassan married Maryam, a Malay-Tamil girl, in Singapore. Hassan was a member of the editorial board of the Singapore Press's newspaper *Utusan Melayu* from 1912 to 1913 ([Mughni 1984](#), p. 12). During this time, Hassan wrote extensively on religious topics. He wrote a lot of moral advice, mostly in the form of poetry, encouraging people to do good and stay away from crime. At the same time, the influence of Islamic renewal ideas he read from *al-Manār* magazine published by Muhammad Rashid Riḍā in Egypt and *al-Imām* magazine published by young Minangkabau scholars is one of the reasons Hassan also criticized the decline of the Muslim world.

Hassan's involvement in Indonesia began around 1921, when he moved to Surabaya to trade and manage his uncle, Abdul Latif's, shop. The uncle recognized that Hassan's harsh criticism had frightened the government. During his stay in Surabaya, his uncle warned him not to associate with Faqih Hasyim. Hassan, on the other hand, was a friend of his, as well as a frequent hangout for several figures from the Islamic movements, including Ahmad Syurkatiy, H.O.S. Cokroaminoto, H. Agus Salim, Mas Mansur, H. Munawar Chalil, Soekarno, Muhammad Maksum, Mahmud Aziz, and many others.

After three years in Surabaya, Hassan relocated to Bandung in 1924. He initially went to Bandung to pursue the textile business but instead became acquainted with the founding figures of the Persatuan Islam organization (Federspiel 1996b, p. 24). His political experience led him to become Sukarno's spiritual teacher while the latter was in exile in Flores (Jamil 2008, p. 200). It is clear that reading materials such as *al-Kafā'ah* by Ahmad al-Syurkati, *Bidāyah al-Mujtahid* by Ibnu Rushd, *Zād al-Ma'ād* by Ibnu Qayyim, *Nayl al-Awtar* by Muhammad 'Alī al-Shawkānī, and *Subul al-Salām* by al-Ṣan'ānī influenced his progressive ideas (Mughni 1984, p. 20). Finally, Hassan chose to live in Bangil beginning in 1940. He established Pesantren Persis there. Hassan was active in writing articles *for Himayat al-Islam* magazine, which he published until his death on 10 November 1958. Hassan was laid to rest in Sengok Cemetery, Bangil.

Ahmad Hassan was a productive thinker who published his ideas in magazines and books. *The al-Furqan Tafsir Qur'an* was the result of a lengthy writing process. The first part of Tafsir was actually published in 1928. The second part of this commentary was only published in 1941. This second issue also does not cover the entire Qur'an because Ahmad Hassan only finished the translation until surah Maryam. It is unclear what caused this commentary to be halted for so long, as Ahmad Hassan was able to continue writing his tafsir work in 1953, when a businessman named Saad Nabhan was willing to pay the cost of publishing his tafsir.

Three years later, in 1956, *al-Furqan* was then completely written 30 juz'. It was an annotated translation of the Qur'an comprising additional information of tafsir in the form of footnotes. In translating the verses Hassan generally used the literal translation method, i.e., word-by-word translation, except for some vocabularies that are not permitted to be exposed in literal translations. Thus, *al-Furqan*'s interpretive characteristic is actually a composite element of both translation and interpretation at once, but with a smaller portion of tafsir.

Changes in the development of the Indonesian language concerning spelling improvements through the adoption of the Enhanced Spelling (EYD) in 1972 had been one of the motivations for translating and interpreting the Qur'an that continued until the mid-1980s. It was also a compelling reason for Hassan's extended family to reconsider the book's use of the Indonesian language. It is necessary to make some changes to the exposed language in order to keep up with the changes and improvements in the development of the Indonesian language. As a result, the book can be easily read by the current generation of Muslims, thanks to the use of modern linguistic spellings. The *al-Furqan Tafsir Qur'an* 2010 edition is a single volume with 1100 pages plus 90 pages of introductory remarks. This book is a global commentary (*ijmalī*) methodologically, and even without the commentary on the front-page cover, this commentary is similar to the Arabic text of the verses and the most widely circulated tafsir at the moment. This interpretation differs from others in that it identifies the author/compiler who is responsible for the translation process.

Ahmad Hassan is the only name on the book's front cover. The title is *al-Furqan Tafsir Qur'an*, and the ISBN number is 978-602-95064-0-2 (Hassan 2010). In contrast to the style of translation and spelling used in the initial 1956 publication, which still uses Malay, albeit with the old Latin scripts, the new edition of *al-Furqan* is in modern Indonesian, as the team reorganized its translations using new spelling and lingual style. The team in charge of rewriting the book is called The Center Team for Language and Cultural Development (Tim Pusat Pengembangan Bahasa dan Budaya), Al-Azhar University Indonesia (UAI).

The current authors are using the 2010 edition, which is the second reprinting. This new edition's first edition appears to have been published in 2005. In addition to language editing guidelines that preserve the author's original nuance, editing is only completed for a few words that have a slightly different meaning than the prevailing notion of the current Indonesian language (Hassan 2010, p. viii). The publisher also mentioned the writer's testimony in the introductory section that both theme titles and sub-themes mentioned in the verses or groups of verses refer to the Medina edition of *Al-Qur'an dan Terjemahnya* by the Ministry of Religious Affairs. Thus, it appears that the table of contents, which

mentions themes from pages lxx to xc, was arranged according to the chronology of the mushaf, which refers to the same work.

Meanwhile, the new 2010 edition includes a glossary that not only refers to the original Arabic terms, such as *Asbāb an-Nuzul*, *ittaqā*, *mufaṣṣal*, *sā'ah*, and *tafṣīl*, but also Indonesian terms such as arrogant (*sombong*), worldly demeanors (*perhiasan dunia*), and others (Hassan 2010, pp. xxxi–xxxv). The subject is divided into two indexes that are alphabetical in order. *First*, an index titled "Searching Guides of Qur'anic Words" (see "Petunjuk Pencarian Kata dalam Qur'an" by Abdul Qadir Hassan). Because there is no dating for this index, it is unknown when it was compiled. In general, this index is thematic but still very simple, and the compiler testified that the composition refers to Muhammad Fu'ād 'Abd al-Bāqī's *Tafsīl Āyāt al-Qur'an al-Hakīm* (Hassan 2010, pp. xxxvi–xl). In short, the index appears to be an Indonesian translation of the work, as represented by terms such as fairness (*adil*), mandate (*amanat*), wine and gambling (*arak dan judi*), happiness (*bahagia*), good deed (*berbuat baik*), cleanliness (*bersih*), stars (bintang-bintang), wasteful (*boros*), earth and sky (*bumi dan langit*), suicide (*bunuh diri*), and so on.

*Second*, in addition to the simple thematic index by Abdul Qadir Hassan, the book includes a more comprehensive thematic index by Zuhal Abdul Qadir titled "The Pursuit of the Qur'anic Teachings" ("Penelusuran Pokok-pokok Ajaran Qur'an"). This index was created on 5 February 2005, in Jakarta. In his introduction, Zuhal emphasized the significance of the thematic index for non-religious readers of the Qur'an, such as himself. Zuhal emphasized that the indexes that divided the grouping of themes into six main aspects—(1) the principles of faith and deity, (2) Muslims and their worship, (3) sciences, (4) moral principles, (5) societal and economic aspects, and (6) legal and state aspects—are a very useful contribution because it aids many people in understanding the contents of the Qur'an (Hassan 2010, p. xli).

According to the above classification, the index is not arranged alphabetically, as previously completed by Abdul Qadir Hasan, but rather by categorizing themes under aspects of discussion. The first theme presented is "Qur'an, Faith, and God-head", which is then divided into five sections with some sub-derivative themes in each. The five main themes are (1) to know the Qur'an, (2) the principles of faith, with some sub themes such as Faith to the Prophet, Angels, and Book; Faith to the Qada', Qadar, The Unseen, and the Resurrection; and Faith to the Qada', Qadar, The Unseen, and the Resurrection, (3) God's Power and Its Main Characteristics, (4) The Life of the World–The Hereafter, and (5) Man's Relationship with His Lord (Hassan 2010, pp. xlii–xlvi). The index is described in three columns, each of which contains a reference to the names and numbers of the surahs of the Koran, the verse numbers, and a brief description of the verses.

## 4. Conclusions

Based on the foregoing explanation, it is possible to conclude that the discourse on thematic interpretation in the Middle East, particularly Egypt, has had a significant influence on the development of thematic interpretation in Indonesia. The re-publication of certain exegetical works by Indonesian exegetes, who used various forms of thematic approach, such as the addition of tables of contents and the arrangement of thematic indexes of Qur'anic verses, is an observable form of development. It is related to Amīn al-Khūlī's (d. 1966) thought, who was an initiator of thematic interpretation. He underlined the importance of thematic discussion in comprehending Qur'anic perspectives on specific cases. Thematic indexes presented by Mahmud Yunus in the back pages of his *Tafsir Qur'an Karim* in the 1950s contain some embryos of Indonesian thematic interpretation. Along with the improvement of the new Indonesian spellings in the 1973 edition, the attempts at re-editing and renewal of the spelling of the book also include some types of indexing the contents of interpretation.

Thematic indexes display not only religious topics but also economic, scientific, historical, and social issues. A similar reshaping of the *Tafsir al-Bayan* written by Hasbie Ash Shiddieqy (first published in 1966) and *al-Furqan Tafsir Qur'an* written by Ahmad

Hassan has been promulgated by Mahmud Yunus in his republishing of the *Tafsir Qur'an Karim* (first published in 1956). When they republished the books in the early 2000s, they improved the Indonesian spelling of both the brief and global styles of *tafsīr*. In order to accommodate the new thematic trend, both *tafsīr* have added a thematic table of contents and a glossarium of thematic indexes. Furthermore, such an influence has influenced the writing of new tafsir, such as Bachtiar Surin's *Terjemah & Tafsiral-Qur'an 30 Juz Huruf Arab dan Latin* and Oemar Bakry's *Tafsir Rahmat*. Thematic indexes have been presented in both tafsirs in the form of chronological and alphabetical tables of contents and thematic indexes, as shown in the Table 1 below:

**Table 1.** Forms of thematic presentations in Indonesian tafsirs.

| First Published | Re-Printed | Book Titles in Indonesian | Author | Table of *Surah*/Theme | Thematic Index | Additional Indexes |
|---|---|---|---|---|---|---|
| 1950 | 1973 | *Tafsir Qur'an Karim* | Mahmud Yunus | Chronological | Thematic–Random | - |
| 1976 | 1976 | *Terjemah dan Tafsir al-Qur'an 30 Juz huruf arab dan latin* | Bachtiar Surin | Chronological/ Alphabetical | - | - |
| 1981 | 1981 | *Tafsir Rahmat* | Oemar Bakry | Alphabetical Chronological | Thematic–Random | Chronological |
| 1966 | 2002 | *Tafsir al-Bayan* | Hasbi Ash-Shiddieqy | Chronological | Alphabetical Glossary | - |
| 1956 | 2005 | *Al-Furqan Tafsir Qur'an* | Ahmad Hassan | Chronological | Thematic–Random | Alphabetical Glossary |

**Author Contributions:** Conceptualization, J.A. and M.A.S.; methodology, J.A.; software, H.H.U.; validation, J.A., M.A.S. and H.H.U.; formal analysis, J.A.; investigation, H.H.U.; resources, M.A.S.; data curation, J.A.; writing—original draft preparation, J.A.; writing—review and editing, J.A. and M.A.S.; visualization, H.H.U.; supervision, J.A.; project administration, J.A.; funding acquisition, J.A. All authors have read and agreed to the published version of the manuscript.

**Funding:** This research received no external funding.

**Institutional Review Board Statement:** Not applicable.

**Informed Consent Statement:** Not applicable.

**Data Availability Statement:** The data presented in this study are available on request from corresponding author.

**Conflicts of Interest:** The authors declare no conflict of interest.

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
