# Peer review of "Thematic Presentations in Indonesian Qur’anic Commentaries"

_religions, doi:10.3390/rel13020140_

Round 1

Reviewer 1 Report

This is an interesting paper documenting an under-researched area, and thus potentially worthy of publication. But at times it is far too descriptive, and not particularly analytical. To illustrate, do we really need to know the phone number of one of the publishing houses referenced (line 275) or the names of all of Hassan's children (lines 579-80). The main argument of the paper appears to be that Middle Eastern tafsir has shaped Indonesian tafsir. This in unsurprising, if the Indonesian scholars concerned trained in the Middle East. What would make the argument more compelling would be if there were clearer links made between Middle Eastern tafsir and that produced in Indonesia.

Alternatively, the paper may have been written primarily to record how Indoneisan Qur'anic commentaries function - that would also make for an interesting paper, but again, a more analytical and less descriptive approach is needed.

Once this revision has been undertaken, I would also recommend thorough proof reading by a native speaker of English.

Author Response

I deleted the publisher's phone number and address (see line 272). I also deleted the names of all of Hassan's children information (lines 565–566).

As a pioneer of Indonesian interpretation and an inspiration to later Indonesian commentators, Mahmud Yunus demonstrated that he was influenced by the ideas of Egyptian reformers Muḥammad ‘Abduh and Rashīd Ridā. Yunus is also an alumnus of Al-Azhar University, Egypt. (Lines 166-173 and 230-238).

Reviewer 2 Report

The article is interesting, well elaborated and majorly comprehensible; however, I would like to ask to the author to better clarify the meaning of the sentence included in verses: 39-42, that it seems unclear to me, as the verse 130: Since it, In fact it....does not appear a grammatically correct. 

In relation to the main thematic approach, I would suggest to better expand, generally speaking, the reasons for the direct correlation between Egypt and Indonesia Qur'anic Tafsir influence and which is the legacy until today in contemporary Indonesian hermeneutical studies.

It is historically evident that the Egyptian's referring influence is linked to authors that from Amin al-Khuli gave a particular contribution to a rational-thematic approach to the last century interpretation of the holy Qur'an; however, very few information is dedicated in this article about the clear impact that those Tafsir printing and re-printing had on the Indonesian Islam as well as in developing an internal approach to exegesis and its conservative-liberal attitude. 

Author Response

I revised it (lines 39–40). The thematic interpretation method is divided into two parts: 1) interpreting one surah and then giving themes based on the contents of the Qur'anic verse, and 2) interpreting by first determining a topic and then gathering Qur'anic verses related to the subject

As a pioneer of Indonesian interpretation and an inspiration to later Indonesian commentators, Mahmud Yunus demonstrated that he was influenced by the ideas of Egyptian reformers Muḥammad ‘Abduh and Rashīd Ridā. Yunus is also an alumnus of Al-Azhar University, Egypt. (Lines 166-173, as well as 230-238).

Regarding Amīn al-Khuli's influence on Indonesian commentators, one example is Mahmud Yunus's commentary (lines 230–238 and 202-229).

Round 2

Reviewer 1 Report

This is a much improved version of this paper, and once it has been thoroughly proof read, it will be suitable for publication. The argument is now much clearer and the contribution to knowledge therefore more explicit.

To illustrate the need for proof reading:

  • There are instances of where singular / plural agreement is lacking (for example line 120 has "It is" which should (I think) be "they are" and line 157 has "has" which should be "have"
  • There is inconsistency - is it "Qur'an" (as it is the majority of the time) or "Quran" (as it is on, for example, line 215, 262, 295) or even "Koran" as on line 871? Please not this is not an exhaustive list of the inconsistencies - you will need to check properly yourself.
  • When you say "boredom" (lines 445, 453 and 467) do you mean "confusion"?
  • Line 539 "propels" should be "propelled"

This is  not an exhaustive list, but simply indicates the need for a proper proof read, spell check etc before final submission.